# ThermoSlope: A Software for Determining Thermodynamic Parameters from Single Steady-State Experiments

**DOI:** 10.3390/molecules26237155

**Published:** 2021-11-26

**Authors:** Bjarte Aarmo Lund, Bjørn Olav Brandsdal

**Affiliations:** Hylleraas Centre for Quantum Molecular Sciences, Department of Chemistry, Faculty of Science and Technology, UiT The Arctic University of Norway, N9037 Tromsø, Norway; bjorn-olav.brandsdal@uit.no

**Keywords:** thermodynamics, transition state theory, enzyme kinetics, Arrhenius equation, Michaelis–Menten kinetics

## Abstract

The determination of the temperature dependence of enzyme catalysis has traditionally been a labourious undertaking. We have developed a new approach to the classical Arrhenius parameter estimation by fitting the change in velocity under a gradual change in temperature. The evaluation with a simulated dataset shows that the approach is valid. The approach is demonstrated as a useful tool by characterizing the *Bacillus pumilus* LipA enzyme. Our results for the lipase show that the enzyme is psychrotolerant, with an activation energy of 15.3 kcal/mol for the chromogenic substrate para-nitrophenyl butyrate. Our results demonstrate that this can produce equivalent curves to the traditional approach while requiring significantly less sample, labour and time. Our method is further validated by characterizing three α-amylases from different species and habitats. The experiments with the α-amylases show that the approach works over a wide range of temperatures and clearly differentiates between psychrophilic, mesophilic and thermophilic enzymes. The methodology is released as an open-source implementation in Python, available online or used locally. This method of determining the activation parameters can make studies of the temperature dependence of enzyme catalysis more widely adapted to understand how enzymes have evolved to function in extreme environments. Moreover, the thermodynamic parameters that are estimated serve as functional validations of the empirical valence bond calculations of enzyme catalysis.

## 1. Introduction

Michaelis–Menten kinetics, a model of enzyme kinetics developed by Leonor Michaelis and Maud Menten, is still commonly used after over 100 years for reactions involving a single substrate. As shown in Equation (1), it relates reaction velocity to substrate concentration [S], the Michaelis constant ($K_{M})$, which is often used as a proxy for the enzyme’s affinity for its substrate and the enzyme’s maximum rate ($V_{max}$). The $V_{max}$ can also be expressed as the product of the enzyme’s turnover number and the initial enzyme concentration ([E]), which is assumed to be unchanging. While the Michaelis–Menten model applies to many systems, it is not universal, and other models, such as the ping-pong bi-bi, have also found applications [1].
(1)$$v=V_{max}\frac{\left[S\right]}{K_{M}+\left[S\right]}=k_{cat}\left[E\right]\frac{\left[S\right]}{K_{M}+\left[S\right]}$$

The Arrhenius equation (Equation (2)) developed by Svante Arrhenius describes the temperature dependence of reaction rates. The rate constant in the Arrhenius equation can be either the *k_cat_*/$K_{M}$ or the *k_cat_* of the Michaelis–Menten equation, and, in this case, the calculated activation barrier ($-E_{a})$ relates to the rate-limiting step of the reaction catalysed by the enzyme. The pre-exponential factor A gives a measure of the number of collisions between the enzyme and substrate per unit of time. The product of the universal gas constant (R) and the experimental temperature (T) weighs the exponential term to be temperature-dependent. As the temperature increases, the reactant’s average kinetic energy increases, which leads to a higher likelihood of productive collisions between the reactants.
(2)$$k_{cat}=Ae^{\frac{-E_{a}}{RT}}$$

This classical biochemistry experiment has traditionally been performed with either saturating substrate concentrations at a set of different temperatures to achieve the maximum velocity ($V_{max}$) or, in the case of a less soluble substrate, with a range of substrate concentrations with the highest concentrations approaching saturation [2]. It is also possible to use non-saturating concentrations to determine the *k_cat_*/$K_{M}$ ratio [3]. This is a labour-intensive experiment, and, since there have not been too many applications for the thermodynamic parameters of the activation barrier, it is not frequently carried out today. However, with the empirical valence bond theory first described by Nobel laureate Arieh Warshel [4], the thermodynamic parameters have received a renewed use as a means to validate computational studies on enzyme catalysis. By simulating the rate-limiting step at different temperatures, it is possible to calculate computational Arrhenius plots and, based on these, to determine the thermodynamic properties of the activation barrier [5,6,7]. Knowledge of the entropic and enthalpic contributions of the enzyme provides a deeper understanding of how enzymes are tuned during evolution to cope with extreme environments, such as the freezing or boiling point of water.

We propose simplifying the protocol by employing sample changers with gradual heating and collecting velocities as a function of both time and temperature. As the temperature increases, one can expect the rates to increase until a maximum is reached, and the rates decrease as the enzyme takes on an inactive conformation or ultimately unfolds [2]. We have developed an open-source web application that uses a rolling regression to determine *k_cat_* as a function of temperature and which carries out the calculations to determine the thermodynamic parameters from a single experiment with multiple cuvettes under a gradual increase in temperature. To demonstrate the principle, we used a simulated dataset based on integrating the Arrhenius equation to show that the algorithm is capable of reproducing the input parameters. To validate our approach, we characterised the cold-active *Bacillus pumilus* LipA [8] (pLipA) with both the traditional stepwise and gradual increase in temperature using the established chromogenic esterase substrate para-nitrophenyl butyrate (pNPB). Although lipases, in many cases, have complex kinetics due to effects such as interfacial activation [1], minimal lipases, such as the *Bacillus subtilis* LipA, are known to be lidless and without interfacial activation [9]. Furthermore, the Michaelis–Menten kinetic parameters have already been determined for both the *B. subtilis* and *B. pumilus* LipA [8,10]. We also employed the approach on three α-amylases, spanning habitats from psychrophilic to thermophilic, and the results clearly differentiate the three enzymes and provide thermodynamic insight into their temperature adaptation. This also shows that the method is useful in comparing different enzymes.

## 2. Results and Discussion

The ThermoSlope software was able to be used to fully reproduce the input parameters for a simulated dataset, as shown in Figure 1.

As a model system, pLipA appears to be well-suited. The enzyme pLipA was found to express well in *E. coli*, with 15 mg of enzyme purified from 1 L of *E. coli* culture. The standard purification steps with immobilised metal affinity chromatography and ion-exchange chromatography yielded homogeneous samples. Structural studies using NMR and X-ray protein crystallography are underway, enabling its use in computational studies as a model system for biocatalysis. To ensure that the protein was well-folded and stable over the studied temperature ranges, differential scanning calorimetry was used to assess the temperature stability of pLipA. A single transition was observed corresponding to a T_m_ of 39.6 °C or 313 K (Appendix A). No refolding was observed, and the pLipA precipitated during the heating cycle, so the fitted ΔH of 447 kcal/mol may have several components, including aggregation, in addition to the folding enthalpy.

With the esterase substrate para-nitrophenyl butyrate, apparent steady-state behaviour was observed for pLipA. Based on these observations and the previous use of Michaelis–Menten kinetics for pLipA and its *B. subtilis* homolog, we decided to use pLipA as the system to compare our new gradient-based approach to the traditional stepwise approach.

Based on Akaike’s information criteria (AICc) [11], a global fit of a single curve (AICc = −37.99, wAICc = 0.997), y = −7712x + 32.06, was found to describe the two datasets (Figure 2, Table 1), deeming the gradient approach (AICc = −26.44, wAICc = 0.003) equivalent to the classical stepwise approach (AICc = −13.36, wAICc = 4.47 × 10^−6^). The linear regression results can be used to estimate the thermodynamic parameters following the established methods [12]. Based on the results, we calculate an $E_{a}$ of 15.3 kcal/mol, $\Delta G^{‡}$ of 13.8 kcal/mol, $\Delta H^{‡}$ of 14.7 kcal/mol and $T\Delta S^{‡}$ of 1 kcal/mol. These values for the activation energy are higher than for other psychrophilic esterases but lower than for typical mesophilic esterases [13]. We have thus shown that pLipA yields equivalent Arrhenius curves using classical stepwise and our new gradient approach.

The motivation for the study was to better understand the thermodynamic underpinnings of enzymes’ adaptations to extreme environments, so we used the ThermoSlope software to evaluate three α-amylases. The α-amylases came from psychrophilic (*Pseudoalteromonas haloplanktis*, AHA), [14] mesophilic (*Sus scrofa*, PPA) and thermophilic (*Bacillus licheniformis*, BLA) hosts and would thus sample different temperature extremes. In Figure 3, we show that Thermoslope gave good results for the three enzymes. There is a clear trend in decreasing Δ*G*^‡^ from the thermophilic to psychrophilic amylase (Table 2), with the psychrophilic AHA having the lowest Δ*G*^‡^ even though it has a higher Δ*H*^‡^ than BLA. It follows that the AHA enzyme accomplishes this by reducing *T*Δ*S*^‡^ to near zero. There are few published studies using 2-Chloro-4-nitrophenyl-α-d-maltotrioside (CNP-G3) with purified enzymes; however, a *k_cat_*-value of 2.98 s^−^^1^ has been reported for the human pancreatic α-amylase at 30 °C [15]. This is on the same order of magnitude as AHA (5 s^−^^1^) and PPA (0.2 s^−^^1^), and, as expected, the thermophilic BLA has significantly lower activity (0.002 s^−^^1^) extrapolating from the Arrhenius curve.

The significance of the new gradient approach is the much-simplified workflow and reduced labour use. Whereas a classical cuvette-based Arrhenius plot would require six concentrations, at least two times duplicated, over at least four different temperatures (nearly 50 measurements), the new approach only needs six concentrations in two or more replicas (for a total of 12 measurements). Even with a sample changer, the reduced effort is significant, not more than the constant temperature Michaelis–Menten steady-state assay requires.

The software (Figure 4) is implemented to support the qChanger6 (Quantum Northwest Inc.) for the Agilent Technologies Cary 60 instrument. However, the underlying architecture is flexible and should accommodate any data source that gives parseable datafiles with time, temperature and absorbance. The code is made available under an open-source license and with a publicly accessible web interface. The software is designed to be deployable to internal infrastructure for confidential data.

The setup of the experiment does require some familiarity with the model system of interest. The range of temperatures chosen depends on the physical properties of the system, including the solubility of the substrate at lower temperatures and stability at higher temperatures. It is also essential to avoid temperatures where the enzyme will be prone to aggregation, reducing the effective enzyme concentration [2]. Buffers have varying degrees of temperature dependence, and, if the pH is believed to be crucial for the activity, it is essential to ensure that the pK_a_ of the buffer system does not change significantly over the temperature range explored. We had differential scanning calorimetry results (Figure 1) for this system to ensure that no unfolding would occur within the temperature range sampled. The temperature ramp should be slow enough that the enzyme equilibrates with the temperature. Furthermore, the temperature ramp rate should be set low enough to maintain enough sampling of absorbances to give reasonable rate estimates while high enough to ensure that the reactant stationary assumption is not violated during the experiment [16]. The scan rate was set to achieve a sampling of each cuvette with every 1 K temperature increase in the experiments presented here. The qChanger6 (QNW) allows for ramping rates spanning from 0.1 to 10 K/min for temperatures between 260 and 380K, and the software makes no assumptions on the rate or temperature. In the current implementation, $K_{M}$ and *k_cat_* are determined for each temperature bin without constraints based on the other bins, and the software allows the researcher to change the temperature cut-offs to eliminate datapoints where the enzyme is no longer active. The number of bins will also be empirically determined by the researcher based on how many datapoints were obtained and how quickly the velocities change throughout the gradient. The fitting is performed straightforwardly, and there are several possible extensions. One would be to fit the curves globally [17] or to use approximate Bayesian computation (ABC) techniques to reduce the number of necessary samples further [18]. There are also arguments for fitting the data to *k_cat_* and *k_cat_*/$K_{M}$ instead, which would remove the need for high concentrations of substrate and reduce the number of samples needed [3].

## 3. Materials and Methods

### 3.1. Software Implementation

ThermoSlope was implemented in Python with the Flask framework for easy deployment to in-house infrastructure or cloud offerings, such as Azure App-services. It uses the pandas [19] package to treat the data files and manipulate the data structures. Furthermore, statsmodels [20], scipy [21], numpy [22], scikit-learn [23] and matplotlib [24] are used for the calculations. The software starts by reading in the long-form CSV file produced by the software that accompanies the qChanger6 software for the Cary60 instrument. Time is converted to seconds, and temperatures are converted to absolute temperatures (Kelvin). Concentrations are calculated from the measured absorbances and the given extinction coefficients. The key step is the rolling regression analysis that calculates reaction rates (velocities) from the concentration against time. A rolling ordinary least squares fitting algorithm from statsmodels [20] is used. A window size of 4 was chosen to minimise the temperature-dependent change in velocity for each window. The rolling regression scheme means that the terminal temperatures of the gradient are not included. The calculated velocities are then binned into a user-selectable number of bins. By increasing the number of bins, there will be less variation in the velocities due to temperature change; however, there will also be fewer observations and possibly worse statistics. For each temperature bin, an average temperature is calculated and Michaelis–Menten kinetics are determined for this temperature using non-linear regression. The non-linear least squares regression is implemented in scipy [21] with the Levenberg–Marquardt algorithm, and the velocities are fitted against the substrate concentrations with the parameters $K_{M}$ and *k_cat_*. Errors are also estimated for the non-linear regression. A weighted least-squares linear regression function from statsmodels [20] is used to calculate the fit of ln (*k_cat_*) to the inverse temperature, with the weights being the inverse of the squared standard error of the *k_cat_*-value.

### 3.2. Recombinant Protein Production

The DNA sequence encoding residues 35–215 of the mature *Bacillus pumilus* Lipase L5 (UniProtKB W8FKE7) was optimised for *E. coli* expression, synthesised and subcloned into pET-22b(+) within the NdeI/XhoI sites by GenScript Biotech (Leiden, The Netherlands). The plasmid was transformed into NiCo21(DE3) chemically competent *E.coli* cells (New England BioLabs, Ipswich, MA, USA) using a standard heat shock protocol. Cells were grown in shake flasks in ZYP-5052 auto-induction media [14] for 4–6 h at 37 °C before the temperature was lowered to 17 °C for overnight expression. Cells were harvested by centrifugation at 7500× *g* for 45 min at 4 °C. Pellets were resuspended in 50 mM HEPES pH 7.5 supplemented with 500 mM NaCl (buffer A) and sonicated for 15 min at a maximum of 12 °C. The cell lysates were clarified by centrifugation at 50,000× *g* for 45 min at 4 °C and filtered through 0.45 µm syringe filters. The clarified lysate was loaded on HisTrap FF 1 mL crude columns on an äkta Pure system (Cytiva, Marlborough, MA, USA) equilibrated with buffer A. Protein was eluted by an increase in imidazole concentration from 50 mM to 375 mM. The eluted protein was desalted using a HiPrep 26/10 Desalting column equilibrated with 20 mM HEPES pH 7.5 (buffer B). The desalted protein was loaded on a HiTrap Q 1 mL column equilibrated with buffer B. The pure protein was obtained by a gradient towards 1 M NaCl. Monodispersity was determined by analytical size exclusion chromatography using a Superdex 75 10/300 column in PBS buffer, and protein identity was confirmed by tandem mass spectrometry. The *Pseudoalteromonas haloplanktis* α-amylase (UniProtKB P29957) was produced following the same protocol but with a cation-exchanger HiTrap SP 1 mL column instead of the anion-exchanger HiTrap Q.

### 3.3. Experimental Setup

The activity of pLipA was measured colourimetrically by following the breakdown of the ester para-nitrophenyl butyrate at 405 nm based on an established protocol [25]. Para-nitrophenyl butyrate (Sigma-Aldrich St. Louis, MO, USA) dissolved in acetonitrile was diluted with 50 mM potassium phosphate pH 7.2 and 5% acetonitrile for the measurements. Independent replicates of dilution series with 1:1 dilution and 1 nM enzyme were used to determine K_m_ and *k_cat_* in a 2 K/min temperature gradient using a Cary60 (Agilent Technologies, Santa Clara, CA, USA) with a Qchanger6 (Quantum Northwest Inc, Liberty Lake, WA, USA) temperature-controlled cuvette changer. Solutions (900 µL) were equilibrated at the lowest temperature for 5 min before starting the reaction with the addition of chilled enzyme stock solution (100 µL, 10 nM concentration) using a multichannel pipette. To ensure mixing, magnetic stir bars for cuvettes were used during the experiments at 1200 rpm. Substrate concentrations were calculated from the dilution factor and the amount of produced product based on an extinction coefficient of 17,800. The stepwise comparison was made in the same manner but with the temperature kept constant at 277.15, 283.15 K, 288.15 K and 293.15 K. The Arrhenius plots were evaluated with Graphpad Prism v6.0 to decide if there were any significant differences between the two approaches.

The activities of the α-amylases were completed in a similar fashion using the ThermoSlope-approach with the chromogenic substrate 2-Chloro-4-nitrophenyl-α-d-maltotrioside (Sigma-Aldrich, cat.no. 93834, St. Louis, MO, USA) in phosphate buffered saline. The reaction was followed by measuring absorbance at 405 nm, with an extinction coefficient of 12,900 M^−1^cm^−1^ being used. *Bacillus licheniformis* α-amylase (BLA, Sigma-Aldrich, cat. no.A3403, St. Louis, MO, USA) was used at a final concentration of 3.8 µM in the temperature range of 303 to 363 K with a ramp of 2 K/min. Pig porcine α-amylase (PPA, Sigma-Aldrich, cat.no. A4268, St. Louis, MO, USA) was used at a final concentration of 0.45 nM between 283 and 363 K with a ramp of 1 K/min. Finally, the *Pseudoalteromonas haloplanktis* α-amylase (AHA) was used at 60 nM concentration between 283 and 306 K with a temperature gradient of 1 K/min.

### 3.4. Differential Scanning Calorimetry

Differential scanning calorimetry was used to assess the thermal stability of pLipA. The purified enzyme was diluted to 1 g/L concentration and dialysed into 50 mM potassium phosphate 7.2. All samples were filtrated and degassed. Temperatures were scanned in the range 25–55 °C with a gradient of 1 °C·min^−1^. All measurements were collected using a CSC Nano-Differential Scanning Calorimeter III (N-DSC III) with constant pressure at 3 atm. All data were analysed in NanoAnalyze 3.6 (TA Instruments, New Castle, DE, USA).

## Figures and Tables

**Figure 1 molecules-26-07155-f001:**
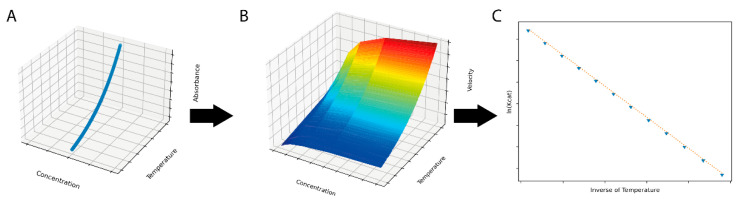
A simulated dataset (**A**) with individual observations generated by integrating the Arrhenius equation over time and temperature was used for input for the ThermoSlope software. The ThermoSlope software uses a rolling regression to estimate the velocities over temperature and time gradients (**B**) and determines the thermodynamic parameters of the activation barrier via the Arrhenius plot (**C**).

**Figure 2 molecules-26-07155-f002:**
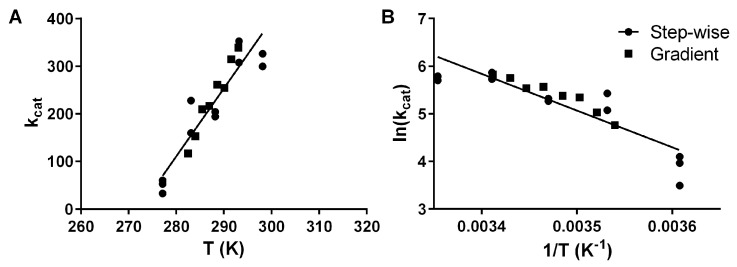
Arrhenius plot of activation energy for the hydrolysis of pNPB by pLipA with either the traditional stepwise approach, where separate experiments are performed for each temperature (circles), or by a gradient approach as we are proposing (squares) as *k_cat_* plotted against the absolute temperature (**A**) and ln(*k_cat_*) plotted against the inverse temperature (**B**). For both plots, the globally fitted linear regression line is shown.

**Figure 3 molecules-26-07155-f003:**
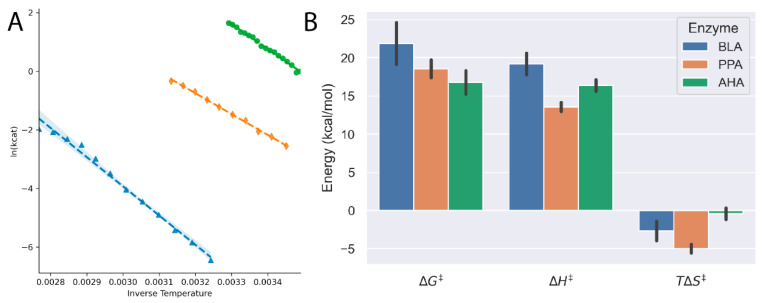
(**A**) Arrhenius plots for *Bacillus licheniformis* amylase (BLA, blue triangles), pig porcine amylase (PPA, orange diamonds) and *Pseudoalteromonas haloplanktis* amylase (AHA, green circles) demonstrate the linear temperature dependence on amylase activity. (**B**) The thermodynamic parameters of the activation barrier show clearly that the cold-adapted AHA has a lower energy barrier to overcome for the reaction to proceed.

**Figure 4 molecules-26-07155-f004:**
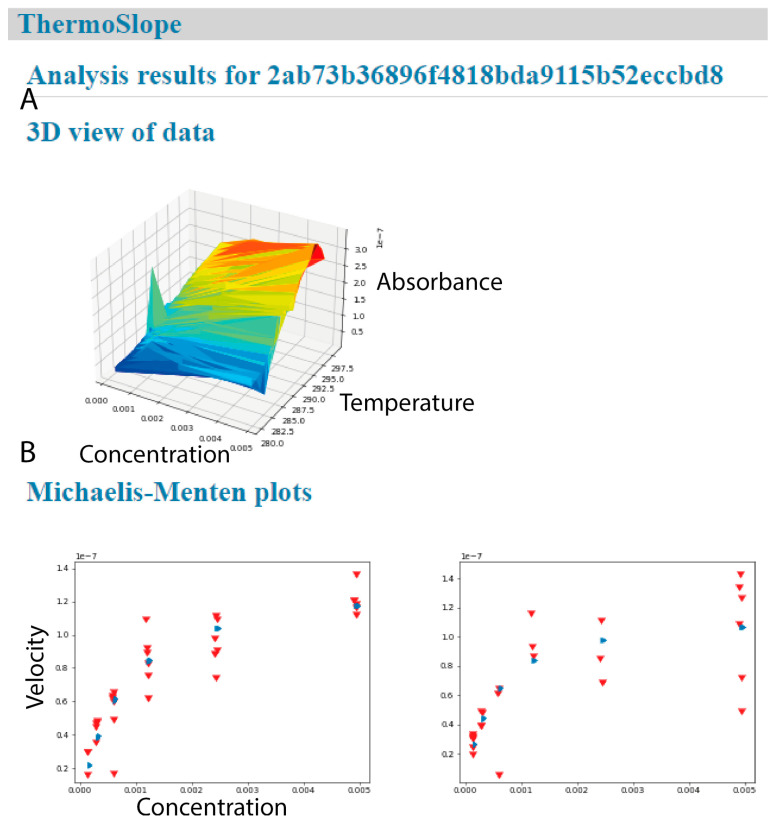
ThermoSlope has a clean interface with modern graphics. The analysis output page gives (**A**) a 3D graphical representation of an example dataset (**B**) and classical Michaelis–Menten plots for each binned temperature range with observed velocities (red) and fitted values (blue).

**Table 1 molecules-26-07155-t001:** Steady-state Michaelis–Menten parameters for pLipA with the substrate para-nitrophenyl butyrate determined using a traditional stepwise approach and our new gradient approach (ThermoSlope).

Temperature (K)	$K_{M}\;(µM)$		*k_cat_* (s^−1^)	
	Stepwise	Gradient	Stepwise	Gradient
277	400 ± 400		50 ± 10	
281		250 ± 50		132 ± 7
282		160 ± 60		110 ± 10
283	1300 ± 800		190 ± 50	
284		200 ± 40		149 ± 9
285		250 ± 50		200 ± 10
287		180 ± 70		200 ± 30
288	500 ± 200		200 ± 20	
289		260 ± 40		250 ± 10
290		180 ± 20		240 ± 10
292		270 ± 60		300 ± 20
293	300 ± 100	240 ± 40	330 ± 40	320 ± 20
295		290 ± 60		370 ± 20
296		210 ± 30		360 ± 20
298	500 ± 100	120 ± 20	310 ± 20	312 ± 10

**Table 2 molecules-26-07155-t002:** Thermodynamic parameters of the activation barrier to the enzymes BLA, PPA, AHA and pLipA given in kcal/mol as calculated by the ThermoSlope software with standard errors calculated from the confidence intervals of the Arrhenius plot.

	BLA	PPA	AHA	pLipA
$\Delta G^{‡}$	21.8 ± 0.2	18.556 ± 0.005	16.782 ± 0.005	13.83 ± 0.06
$\Delta H^{‡}$	18 ± 1	13.5 ± 0.3	16.3 ± 0.4	15 ± 2
$T\Delta S^{‡}$	−4 ± 1	−5.0 ± 0.3	−0.5 ± 0.4	1 ± 2

## Data Availability

All datasets used in the studiy are available from https://github.com/bjartelund/Thermoslope (10.5281/zenodo.5726675).

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
