# Peer review of "ThermoSlope: A Software for Determining Thermodynamic Parameters from Single Steady-State Experiments"

_molecules, 2021, doi:10.3390/molecules26237155_

Round 1

Reviewer 1 Report

The revised form of the manuscript entitled "ThermoSlope: a software for determining thermodynamic pa-2 rameters from single steady-state experiments", can be published in the Journal Molecules.

Author Response

We are very pleased that Reviewer 1 found that our revisions were adequate.

Reviewer 2 Report

Lund and colleagues have done a significant effort to improve their manuscript, and I think that it is now suitable for publication in Molecules after minor revisions.

Minor 1:

Figure 3A lacks of any scale in the graph. The reader cannot understand directly from the graph which are the ranges of measurement for each enzyme, not even the ln(kcat) could be found. 

Minor 2:

Figure 3B should feature thermodynamic values for the activated complex. The symbols adopted should be used accordingly.

Minor 3:

According to reference 12, error should be reported for the thermodynamic parameters. Therefore I suggest to report the error on the bar graph in figure 3B and to report it also in table 2. It depends directly from kcat evaluation. 

Minor 4:

the authors may try to compare their kcat values as calculated at 25°C to the reported kcat for the same enzymes as evaluated in the literature.

Minor 5:

references are not always reported in the same and right style.

Author Response

We appreciate the thorough work of the reviewers and have worked with the manuscript to address the points raised by reviewer 2.

Minor 1, Figure 3A has been revised to include scales on X-axis and Y-axis. 

Minor 2, Figure 3B has been revised with the proper symbols for the activated complex. 

Minor 3, Figure 3B has been amended with error-bars corresponding to the confidence intervals determined for the Arrhenius plot. Furthermore, Table 2 has been extended with the calculated standard errors.

 Minor 4, It has been challenging to find reports on purified enzymes with this substrate. Most reports are with serum-samples where no enzyme concentration is given and thus no kcat can be calculated. We did eventually find a study, which is now cited in the text, where turnover-numbers on the same order of magnitude were observed at 30°C. 

Minor 5, Some of the software citations are proceedings and not published papers which complicates the references. We have revised some journal abbreviations and hope that this will be satisfactory.

This manuscript is a resubmission of an earlier submission. The following is a list of the peer review reports and author responses from that submission.

Round 1

Reviewer 1 Report

I appreciated the effort from the authors in order to improve the overall soundness of the paper, but some open questions remain and I have to reject the paper in the present form.

In details, the author's choice in binning the temperature in that way is very arbitrary and not generalizable. How much the calculation would have changed by binning temperature differently? How a general user can choose the right binning of his data? 

Michaelis Menten plots are generally built from starting rates, taken at time zero, this is done to be more independent from substrate concentration decrease and sometimes enzyme degradation during turnovers. How the authors comment on that? We are not following starting rates, but we are actually binning systematically different conditions of substrate concentration and possibly enzyme "viability".

More in general Michaelis Menten treatment is true as long as the steady state approximation is valid, this then provides an apparent thermodynamic constant being Km. This means that Km changes with temperature as well. Are the authors implying that in their temperature range Km does not change? If yes they should state this very clearly and say that their approach is not generalizable in those cases where Km changes significantly with temperature.

Overall I suggest the authors two possible ways to publish their work that I still find interesting, but in the present form not ready to be published. Either the authors may provide a clear theoretical demonstration of their approach or the authors should provide at least other two benchmark tests with different enzymes. I would suggest to choose three enzymes from three different sources, possibly from psychro-, meso-, thermophilic, respectively.

Reviewer 2 Report

This manuscript reports interesting data and results. Nevertheless authors chosen an enzyme whose kinetic behavior cannot be described accurately by a simple (Michaelis-Menten) model equation. It ashonish me that although authors know the previous truth, as it can be concluded for their text, however have proceed to present a new concept on the temperature dependence of enzyme catalysis, by developing a new approach vs. those of the classical Arrhenius parameter estimation, using a lipase enzyme (indepedently that it is lid-less).

The previous comment is more or less obvious by a simple inspection of Figure 2.

 I have two more remarks:

1) The evaluation of fitting procedures through Akaike’s information criterion can be improved by using also Akaike’s weights. I strongly suggest to authors the use of these criteria.

2) I may conclude that the pLipA activity measurements (colourimetrically) were performed after the inactivation of Lipase through the alkalification of the reaction solution; otherwise the color cannot be appeared at a pH-value 7.2. Such non-continuous assay methods add errorneouss measurements; these errors can be minimized by performing replicate measurements (< 8) in order to avoid, also, outliers.

  Therefore, I suggest a minor revision of the manuscript before its publication, according to the above commens and suggestions.